# Object-aware lifting for
# 3D scene segmentation in Gaussian splatting

## Abstract

Lifting is an effective technique for producing a 3D scene segmentation by un-projecting multi-view 2D instance segmentations into a common 3D space. Existing state-of-the-art lifting methods leverage contrastive learning to learn a feature field, but rely on a hyperparameter-sensitive and error-prone clustering post-process for segmentation prediction, leading to inferior performance. In this paper, we propose a new unified *object-aware lifting* approach in a 3D Gaussian Splatting field, introducing a novel learnable *object-level codebook* to account for objects in the 3D scene for an explicit object-level understanding. To start, we augment each Gaussian point with an additional Gaussian-level feature learned using a contrastive loss. More importantly, enabled by our object-level codebook formulation, we associate the encoded object-level features with Gaussian-level point features for segmentation predictions. Further, we design two novel modules, the association learning module and the noisy label filtering module, to achieve effective and robust codebook learning. We conduct experiments on three benchmarks, *i.e.*, LERF-Masked, Replica, and Messy Rooms datasets. Both qualitative and quantitative results manifest that our new approach significantly outperforms the existing methods in terms of segmentation quality and time efficiency.

## 1 Introduction

Accurate 3D scene segmentation enhances scene understanding and facilitates scene editing, benefiting many downstream applications in virtual reality, augmented reality, and robotics. However, accurate 3D scene segmentation is challenging to obtain, due to limited 3D dataset size and labor-intensive manual labeling in 3D. To bypass these challenges, recent studies (Zhi et al., 2021; Siddiqui et al., 2023; Bhalgat et al., 2023) suggest lifting 2D segmentations predicted by foundation models (Kirillov et al., 2023; Cheng et al., 2022) to the 3D scene modeled by a radiance field for instance-level understanding. Yet, 2D instance segmentations predicted by models like SAM (Kirillov et al., 2023) lack consistency across different views, *e.g.*, the same object may have different IDs when viewed from different angles, leading to conflicting supervision. Besides, inferior segmentations, *e.g.*, under- or over-segmentation, also make the lifting process challenging.

Various strategies have been proposed to address the above issues. An early work Panoptic Lifting (Siddiqui et al., 2023) trains a NeRF to render instance predictions and matches the model's 3D predictions with the initial 2D segmentation masks (see Fig. 1 (a)). However, its learned NeRF representation lacks semantically meaningful instance features to effectively represent objects, thus limiting its performance. Subsequently, (Ye et al., 2023; Lyu et al., 2024) propose object association techniques as a preprocessing to prepare view-consistent 2D segmentation maps with improved multi-view consistency (see Fig. 1 (b)). However, the preprocessing stage often struggles to produce accurate results and the accumulated error can further degrade the performance. The recent state-of-the-art methods (Bhalgat et al., 2023; Ying et al., 2024) encode instance information in the feature field using contrastive learning and apply a clustering as a postprocessing to produce the final segmentations (see Fig. 1 (c)). Though significant improvements are achieved, without a global object-level understanding across different views, their segmentation capability is still bounded. Moreover, their performance is always constrained by the naive clustering postprocess, which is hyperparameter-sensitive and also induces error accumulation. Given the above concerns, we come up with this question: *"Can we have a unified lifting framework by incorporating an explicit object-level understanding for accurate 3D scene segmentation, without pre- or post-processing?"*

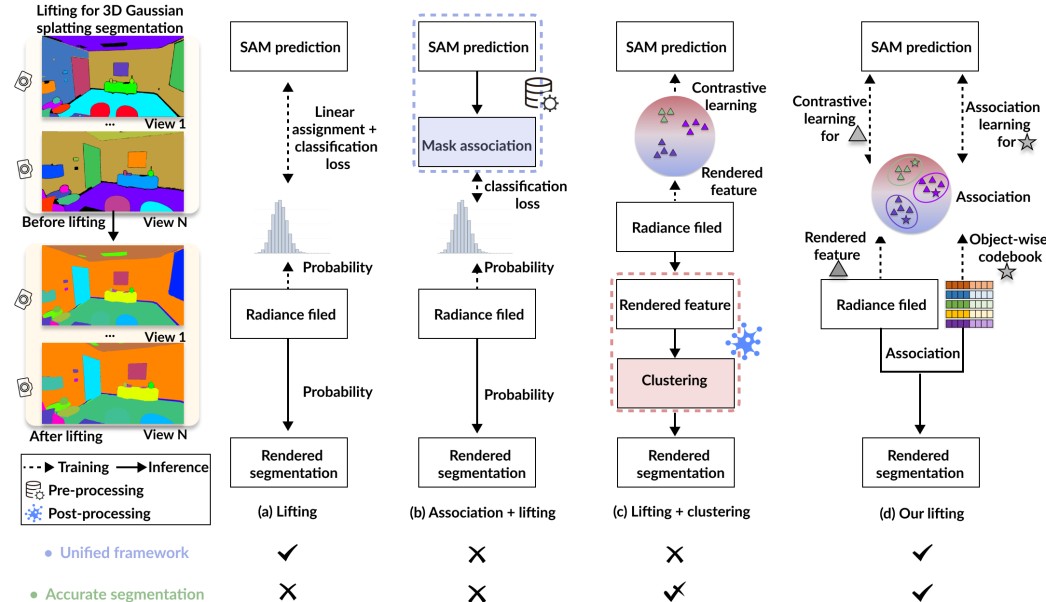

Figure 1: Comparing the pipeline of our method against previous lifting solutions. We refer to the lifting pipeline as a "unified framework" when it does not require pre-processing or post-processing.

In this work, we propose a new unified *object-aware* lifting pipeline for accurate 3D scene segmentation, facilitating the generation of coherent and view-consistent instance segmentation across different views. We exploit the recent advancement of the radiance field, *i.e.*, 3D Gaussian splatting (3D-GS) (Kerbl et al., 2023), as the 3D scene representation due to its superior efficiency and rendering quality. Basically, we augment each Gaussian point in 3D-GS with a Gaussian-level feature and learn these features using contrastive learning defined in each individual view. In particular, we introduce a novel object-level codebook to represent each object in the 3D scene. This codebook is further associated with the rendered Gaussian-level features to predict segmentation results, enhancing object-level awareness during training. Moreover, we present effective learning strategies to optimize the object-level codebook. First, we introduce a novel association learning module, in which we design an area-aware ID mapping algorithm to generate pseudo-labels of association with enhanced multi-view consistency. Additionally, we present two complementary loss functions, *i.e.*, sparsity and concentration parts, to achieve more reliable object-level understanding. Second, we design a novel noisy label filtering module to enhance the robustness of our method by estimating an uncertainty map for the segmentation masks, leveraging the learned Gaussian-level features in a self-supervised manner. During inference, we obtain novel-view instance segmentation results without any pre- or post-processing, effectively avoiding error accumulation.

To evaluate the effectiveness of our method, we conduct experiments on the widely-used LERF-Masked (Ye et al., 2023) dataset and the indoor scene dataset, Replica (Straub et al., 2019). Both quantitative and qualitative results demonstrate that our method outperforms all the existing lifting methods by a notable margin. Furthermore, we conduct additional experiments on the challenging Messy Rooms dataset (Bhalgat et al., 2023), where each scene contains up to 500 objects, demonstrating the scalability of our method in handling large numbers of objects.

Our main contributions are summarized as follows:

- We propose a new unified *object-aware* lifting pipeline for accurate 3D scene segmentation by introducing an object-level codebook representation.

- We present a novel association learning module and a noisy label filtering module to facilitate effective learning of the object-level codebook.

- We set a new state-of-the-art performance on multiple datasets and demonstrate strong scalability in handling large numbers of objects.

## 2 RELATED WORKS

**Radiance field: from implicit to explict.** Radiance field emerges as a promising representation for reconstructing 3D scenes with various properties, *e.g.*, geometries, colors, and semantics, from only 2D inputs such as RGB images and segmentation masks. Neural Radiance Field (NeRF) (Mildenhall et al., 2021) models the radiance field using a neural network composed of layers of multilayer perceptrons. Since then, various works attempt to improve the efficiency of NeRF, *e.g.*, by explicitly formulating the field using 3D structures such as voxels (Chen et al., 2022; Liu et al., 2020) and hash grids (Müller et al., 2022). Later on, 3D Gaussian Splatting (3D-GS) (Kerbl et al., 2023; Xu et al., 2024; Liang et al., 2024; Zhang et al., 2024b; Cheng et al., 2024; Huang et al., 2024; Yu et al., 2024) is introduced to model the radiance field as a set of explicit Gaussian points. This approach allows for a splatting-style rendering (Kopanas et al., 2021), which is highly efficient and demonstrates great potential of real-time rendering. Given these advantages, we employ 3D-GS as the backbone representation in our framework for creating consistent 3D segmentations.

**Segmentation: from 2D to 3D.** Segmentation is a long-standing task in computer vision research. Recent progress witnesses advancements in 2D, thanks to the availability of large-scale datasets. Notably, various foundation models, such as SAM (Kirillov et al., 2023) and its subsequent works (Xiong et al., 2024; Li et al., 2023), show great performance in numerous 2D segmentation tasks and demonstrate robust zero-shot segmentation capabilities.

Beyond segmenting pixels in image level, 3D segmentation aims to partition 3D structures, such as point clouds and voxels (Zhou et al., 2021; Sirohi et al., 2021; Milioto et al., 2020; Gasperini et al., 2021), or to perform segmentation and 3D reconstruction simultaneously from input 2D images (Dahnert et al., 2021; Narita et al., 2019; Rosinol et al., 2020). However, due to tedious work needed in collecting annotated 3D data, the scale of 3D datasets (*e.g.*, 1,503 scenes in ScanNet (Dai et al., 2017)) is usually at least one order of magnitude smaller than that of 2D datasets (*e.g.*, 11M diverse images and 1.1B high-quality segmentation masks in SA-1B (Kirillov et al., 2023)). Hence, the trained models are applicable mostly to limited 3D object categories within the available dataset. To effectively construct 3D segmentation, we propose lifting the segmentation results from 2D foundation models by explicitly incorporating an object-level understanding of the 3D scene.

**Lifting 2D segmentation to 3D scene understanding in radiance field.** Various works (Zhi et al., 2021; Qin et al., 2024; Bhalgat et al., 2023) propose leveraging radiance fields to lift independently-inferred 2D information into the 3D space for 3D scene segmentation and understanding. Some works focus on semantic segmentation, aiming to infer semantic information in the 3D scene, such as object properties and categories, where 2D segmentation predictions are obtained via differentiable rendering. To accomplish this, most existing works tend to optimize a 3D radiance field, which is supervised by semantic or feature maps derived from 2D foundation models. For example, Semantic-NeRF (Zhi et al., 2021) optimizes an additional semantic field from 2D semantic maps for novel-view semantic rendering. Besides, some studies (Zhang et al., 2024a; Qin et al., 2024; Kerr et al., 2023) distill CLIP (Radford et al., 2021) or DINO (Oquab et al., 2023) features into a feature radiance field to facilitate open-vocabulary semantic segmentation.

Unlike semantic segmentation, instance segmentation predicted by 2D foundation models, such as SAM (Kirillov et al., 2023) and MaskFormer (Cheng et al., 2022), lack consistency across multiple views. An early work, Panoptic Lifting (Siddiqui et al., 2023), formulates the radiance field as a distribution of instance IDs and employs the Hungarian algorithm for each 2D segmentation to obtain pseudo labels as the supervision signal. To improve the performance, later works (Ye et al., 2023; Lyu et al., 2024; Dou et al., 2024) attempt to pre-process the 2D instance segmentations (*e.g.*, using video tracker (Cheng et al., 2023) or heuristic Gaussian matching (Lyu et al., 2024)) to simplify the task and obtain view-consistent labels for supervision. Recent state-of-the-art methods (Bhalgat et al., 2023; Ying et al., 2024; Kim et al., 2024; Choi et al., 2024; Dou et al., 2024) construct 3D consistent feature fields and supervise them using contrastive loss within each 2D segmentation. This avoids the need to establish correspondences between different views. However, since radiance fields contain only features, inferring the final segmentation requires an additional clustering step, such as HDBSCAN (McInnes et al., 2017), which can be rather sensitive to the choice of the hyperparameters. In this work, we propose a new unified *object-aware lifting* pipeline for accurate 3D scene segmentation, avoiding the need of pre- or post-processing. By formulating an object-level codebook representation and designing dedicated modules for effective codebook learning, we obtain an object-level understanding of the scene to greatly enhance the segmentation quality.

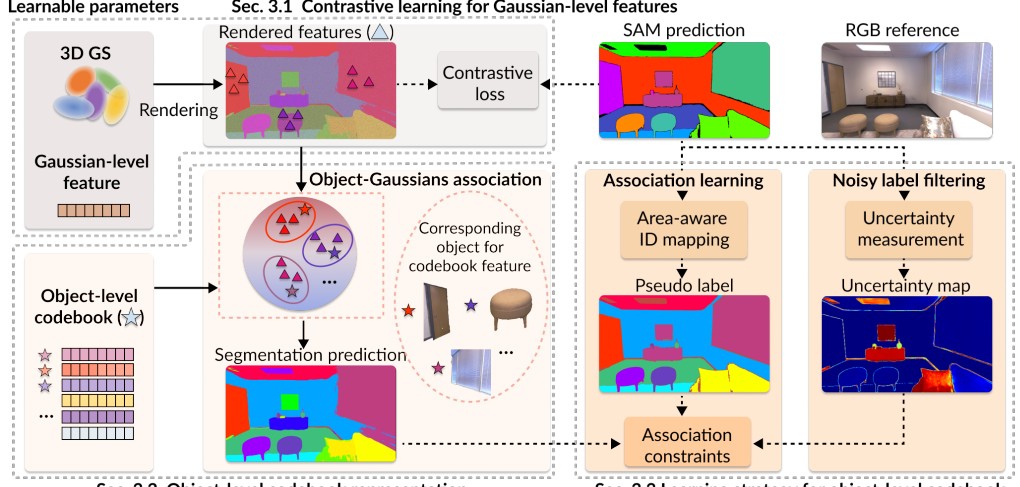

Figure 2: Overview of our unified *object-aware* lifting pipeline, which is built based on the 3D Gaussian Splatting (3D-GS) representation (top-left). In our pipeline, we first augment each Gaussian point in 3D-GS with a Gaussian-level feature and utilize the contrastive loss to optimize the rendered features (see top; detailed in Sec. 3.1). To impose an object-level understanding on the 3D scene, we introduce an additional object-level codebook and establish associations between the object-level features and the Gaussian-level features (see bottom-left; detailed in Sec. 3.2). Further, we propose two novel modules, the association learning module and the noisy label filtering module, to robustly and accurately learn the codebook (see bottom-right; detailed in Sec. 3.3).

## 3 METHOD

Given a set of posed images with 2D instance segmentation masks $\{\mathcal{K}\}$, our goal is to lift 2D segmentations to 3D and produce an accurate and consistent 3D segmentation of the scene, represented by the 3D Gaussian Splatting (3D-GS) model. In this work, we obtain the initial 2D masks using a zero-shot 2D segmentation model, specifically the Segment Anything Model (SAM). Fig. 2 illustrates the overview of our approach, which consists of three major components. (i) We augment each Gaussian point in the 3D-GS representation with an additional Gaussian-level feature and employ contrastive loss to optimize the rendered Gaussian-level features (Fig. 2 top; detailed in Sec. 3.1). (ii) We impose an object-level understanding on the 3D scene to enhance segmentation quality by formulating an object-level codebook and associating the codebook with the Gaussian-level features through an object-Gaussians association for segmentation predictions (Fig. 2 bottom-left; detailed in Sec. 3.2). (iii) We introduce two novel modules for effective codebook learning based on the object-Gaussians association: the association learning module and the noisy label filtering module (Fig. 2 bottom-right: detailed in Sec. 3.3).

### 3.1 PRELIMINARIES

**3D-GS.** The 3D Gaussian Splatting (3D-GS) model (Kerbl et al., 2023) encapsulates a 3D scene using explicit 3D Gaussians and utilizes differentiable rasterization for efficient rendering. Mathematically, 3D-GS aims to learn a set of $N$ 3D Gaussian points $G = \{g_i\}_{i=1}^N$, where $g_i = \{\mathbf{p}_i, \mathbf{s}_i, \mathbf{q}_i, o_i, \mathbf{c}_i\}$ represents the trainable parameters for the $i$-th Gaussian point. The 3D Gaussian function $G_i(x)$ is defined by the center point $\mathbf{p}_i$, the scaling factor $\mathbf{s}_i$, and the quaternion $\mathbf{q}_i$. Moreover, $o_i$ is the opacity value and $\mathbf{c}_i$ is the color values modeled by spherical harmonics coefficients. Following an efficient tile-based rasterization introduced in Kerbl et al. (2023), the 3D Gaussian function $G_i$ is first transformed to the 2D Gaussian function $G_i'$ on the image plane. Then, a rasterizer is designed to sort the 2D Gaussians and employ the $\alpha$-blending to compute the color $\mathbf{C}_u$ for the query pixel $u$: $\mathbf{C}_u = \sum_{i \in \mathcal{N}} \mathbf{c}_i \alpha_i \prod_{t=1}^{i-1}(1 - \alpha_t), \alpha_i = o_i G_i'(u)$, where $\mathcal{N}$ is the number of sorted 2D Gaussians associated with pixel $u$. Subsequently, all parameters in $\{g_i\}_{i=1}^N$ are optimized using the photometric loss between the rendered colors and the observed image colors.

**Contrastive learning for Gaussian-level features.** To encode the instance segmentation information of the 3D scene, each 3D Gaussian point $g_i$ is augmented with a Gaussian-level learnable feature $\mathbf{f}_i \in \mathbb{R}^d$, where $d$ is the feature dimension. Similar to the color information, we can apply differentiable rasterization to efficiently render the feature $\mathbf{F}_u$ for pixel $u$: $\mathbf{F}_u = \sum_{i \in \mathcal{N}} \mathbf{f}_i \alpha_i \prod_{t=1}^{i-1}(1 - \alpha_t), \alpha_i = o_i G_i'(u)$. Following existing state-of-the-art methods (Bhalgat et al., 2023; Ying et al., 2024), we employ the contrastive learning technique to optimize the Gaussian-level features $\mathbf{f}_i$ from individual views. Specifically, we apply the following InfoNCE loss (Li et al., 2020) to supervise the rendered features:

$$\mathcal{L}_{\text{contra}} = -\frac{1}{|\Omega|} \sum_{\Omega_j \in \Omega} \sum_{u \in \Omega_j} \log \frac{\exp\left(\text{sim}\left(\mathbf{F}_u, \overline{\mathbf{F}}_j\right)\right)}{\sum_{\Omega_l \in \Omega} \exp\left(\text{sim}\left(\mathbf{F}_u, \overline{\mathbf{F}}_l\right)\right)}, \tag{1}$$

where similarity kernel function $\text{sim}$ uses the dot product operation here and $\Omega$ is the set of pixel samples. In specific, $\Omega_j$ denotes the pixel samples with the same instance ID $j$ according to the 2D segmentation $\mathcal{K}$, $\overline{\mathbf{F}}_j$ and $\overline{\mathbf{F}}_l$ represent the mean features (centroids) for $\Omega_j$ and $\Omega_l$, respectively.

### 3.2 OBJECT-LEVEL CODEBOOK REPRESENTATION

While Gaussian-level features implicitly encode instance information within the scene, they lack explicit object-level understanding and require an additional clustering post-process to extract this information for segmentation prediction (Bhalgat et al., 2023; Ying et al., 2024). Consequently, these methods not only suffer from tedious hyperparameter tuning but also encounter issues such as under- or over-segmentation due to the accumulated errors (see, *e.g.*, Fig. 3). In contrast, we propose to obtain an explicit object-level understanding of the 3D scene by directly learning from the Gaussian-level features, rather than utilizing a post-processing.

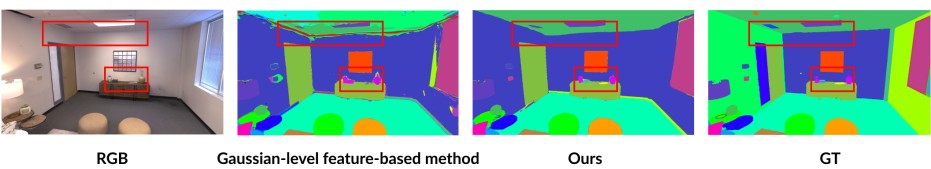

RGB  Gaussian-level feature-based method  Ours  GT

Figure 3: Visual comparisons. Segmentation results produced by our method against the Gaussian-level feature-based method OmniSeg3D-GS (Ying et al., 2024) that uses HDBSCAN (McInnes et al., 2017) post-processing with best-found hyper-parameter. Their result tends to overlook small objects and produces artifacts. In contrast, our method generates more accurate segmentations.

**Object-level codebook.** As shown in Fig. 2 bottom-right, based on the Gaussian-level features, we introduce a learnable object-level codebook representation to impose an object-level understanding of the 3D scene. Practically, we represent the object-level codebook as a compact matrix $\mathbf{F}_{obj} := [\mathbf{F}_{obj}^1, \mathbf{F}_{obj}^2, \cdots, \mathbf{F}_{obj}^L]^T$, where $\mathbf{F}_{obj} \in \mathbb{R}^{L \times d}$, L is the maximum object number, and d denotes the same feature dimension used in the Gaussian-level features. Notably, each row in the matrix $\mathbf{F}_{obj}$ corresponds to an underlying object in the 3D scene.

We further establish the object-Gaussian association formulation to connect the object-level codebook with the Gaussian-level features. Given a pose, we render the feature map $\mathbf{F}$ from the optimized Gaussian-level features, with $\mathbf{F}_u \in \mathbb{R}^d$ denoting the feature for pixel $u$. Particularly, we propose the following association equation to calculate the probability distribution $\mathbf{P}_u \in \mathbb{R}^L$ for pixel $u$:

$$\mathbf{P}_u = \left[\frac{\exp\left(\text{sim}\left(\mathbf{F}_u, \mathbf{F}_{obj}^1\right)\right)}{\sum_{o=1}^{L} \exp\left(\text{sim}\left(\mathbf{F}_u, \mathbf{F}_{obj}^o\right)\right)}, \frac{\exp\left(\text{sim}\left(\mathbf{F}_u, \mathbf{F}_{obj}^2\right)\right)}{\sum_{o=1}^{L} \exp\left(\text{sim}\left(\mathbf{F}_u, \mathbf{F}_{obj}^o\right)\right)}, \cdots, \frac{\exp\left(\text{sim}\left(\mathbf{F}_u, \mathbf{F}_{obj}^{L-1}\right)\right)}{\sum_{o=1}^{L} \exp\left(\text{sim}\left(\mathbf{F}_u, \mathbf{F}_{obj}^o\right)\right)}\right], \tag{2}$$

where we use the same similarity kernel function $\text{sim}$ as in Eq. 1, maintaining consistency with the learning of Gaussian-level features.

**Baseline strategy for learning the object-level codebook.** To automatically learn the object-level codebook during training, a straightforward solution is to directly optimize the object-Gaussians association predictions. To obtain the pseudo-labels for this optimization, we can match the 2D

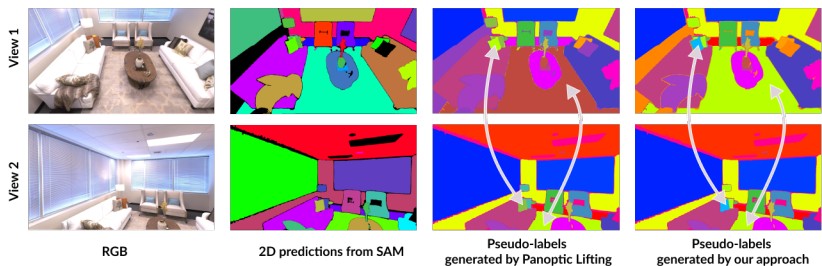

Figure 4: The comparison between the generated pseudo label results by Panoptic Lifting (Siddiqui et al., 2023) and our method. With the designed area-aware ID mapping, we can obtain more view-consistent segmentation as the pseudo labels to facilitate the codebook learning.

segmentation results with the current object-Gaussians association results via the linear assignment algorithm (Kuhn, 1955). In practice, we first need to recover the mapping $\Pi$ from the original instance IDs in the 2D segmentation to the global IDs $\{0, 1, 2, \cdots, L-1\}$ in the 3D scene. Following Siddiqui et al. (2023), the expected mapping $\Pi^\star$ is defined by:

$$\Pi^\star := \underset{\Pi}{\operatorname{argmax}} \sum_{\Omega_j \in \Omega} \sum_{u \in \Omega_j} \frac{\mathbf{P}_u\left(\Pi(j)\right)}{|\Omega_j|}, \tag{3}$$

where $\mathbf{P}_u\left(\Pi(j)\right)$ is the $\Pi(j)$-th value in the probability prediction $\mathbf{P}_u$. Then, we apply the cross-entropy loss as a sparsity term to regress the probability distribution based on calculated pseudo-labels:

$$\mathcal{L}_{\text{class}} := -\frac{1}{|\Omega|} \sum_{u \in \Omega} \log \mathbf{P}_u\left(\Pi^\star\left(\mathcal{K}_u\right)\right), \tag{4}$$

where the $\mathcal{K}_u$ is the instance ID for pixel $u$, given the 2D instance segmentation masks $\mathcal{K}$.

**Inference with the object-level codebook.** Benefiting from the learned explicit object-level codebook representation, our method achieves an end-to-end segmentation inference without the need for a complicated post-processing. In general, to render a segmentation in novel views, we first (i) render the Gaussian-level features; then (ii) calculate the probability using the object-Gaussians association equation; and (iii) determine the segmentation ID by selecting the index of the codebook that exhibits the highest similarity. Furthermore, the same association equation can be directly applied to determine the instance ID for each 3D Gaussian.

### 3.3 LEARNING STRATEGY FOR OBJECT-LEVEL CODEBOOK

Although our baseline strategy for learning the codebook is technically feasible, it faces limitations in terms of performance and robustness. To address these challenges and improve codebook learning, we introduce two novel modules: the association learning module and the noisy label filtering module.

#### 3.3.1 ASSOCIATION LEARNING MODULE

Our association learning module aims to improve the multi-view consistency of pseudo-labels and provide more robust association constraints. To achieve this, we introduce an area-aware ID mapping method and a concentration term to ensure more comprehensive association constraints.

**Area-aware ID mapping.** We observe that the ID mapping described in Eq. 2 is sensitive to the small segments in specific views, thereby further causing the multi-view inconsistency issue, as shown in Fig. 4, To mitigate this issue and improve the multi-view consistency of the generated pseudo-labels, we propose an area-aware ID mapping function, formulated as:

$$\Pi^\star := \underset{\Pi}{\operatorname{argmax}} \sum_{\Omega_j \in \Omega} \sum_{u \in \Omega_j} \mathbf{P}_u\left(\Pi(j)\right). \tag{5}$$

Compared to the previous formulation in Eq. 3, the key distinction lies in the removal of the normalization term. This design prioritizes the influence of large segments in the mapping process, resulting in more consistent mapping across views, as qualitatively shown in Fig. 4. More analysis is provided in Sec. 4.4 and the supplementary material.

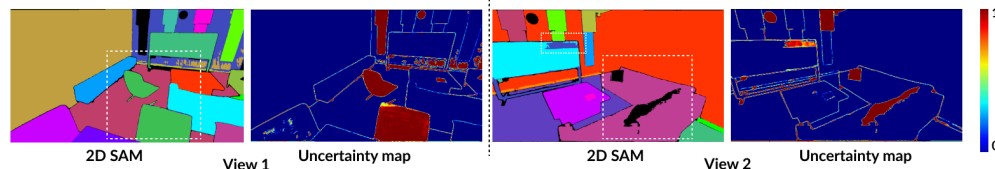

**2D SAM**    View 1    **Uncertainty map**      **2D SAM**    View 2    **Uncertainty map**

Figure 5: Visual comparison of the generated uncertainty maps and 2D instance segmentation masks from different views from the "Office3" scene in the Replica dataset (Straub et al., 2019).

**Concentration term.** We assume that the object-level features assigned in the codebook should align with the clustered Gaussian-level features. Moreover, the clustered Gaussian-level features, optimized using a contrastive loss with dot product similarity, tend to exhibit similar directions. Building on this insight, we propose an additional concentration constraint to minimize the directional differences between the codebook and all corresponding normalized Gaussian-level features:

$$\mathcal{L}_{\text{concen}} := \frac{1}{|\Omega|} \sum_{u \in \Omega} \|\mathbf{F}_{obj}^{\Pi^\star(\mathcal{K}_u)} - \mathbf{F}_u/\|\mathbf{F}_u\| \|_1. \tag{6}$$

Thus, we formulate the total association constraint loss as a linear combination of the sparsity component in Eq. 4 and the concentration component in Eq. 6, providing a comprehensive association constraint for the object-level codebook.

### 3.3.2 NOISY LABEL FILTERING MODULE

To enhance the robustness against noise in the 2D instance segmentation masks, we propose a filtering module that removes less accurate 2D predictions by leveraging multi-view consistently rendered Gaussian-level features. Specifically, we calculate the uncertainty value $\mathbf{W}_u$ for pixel $u$ as

$$\mathbf{W}_u = 1 - \frac{\exp\left(\text{sim}\left(\mathbf{F}_u, \overline{\mathbf{F}}_{(\mathcal{K}_u)}\right)\right)}{\sum_{\Omega_l \in \Omega} \exp\left(\text{sim}\left(\mathbf{F}_u, \overline{\mathbf{F}}_l\right)\right)}, \tag{7}$$

where $\overline{\mathbf{F}}_{(\mathcal{K}_u)}$ is the mean feature (centroid) for $\Omega_{(\mathcal{K}_u)}$. In practice, we model the uncertainty by assessing whether the features corresponding to the current 2D instance segmentation are sufficiently discriminative. Accordingly, we can effectively filter out labels with high uncertainty values (*i.e.*, noisy labels) and integrate this filtering into our association constraints. The overall loss for our proposed learning strategy of the object-level codebook is

$$\mathcal{L} = -\frac{1}{|\Omega|} \sum_{u \in \Omega} \mathbf{1}_{(\mathbf{W}_u \leq \tau)} \left( \underbrace{w_{\text{class}} \log \mathbf{P}_u \left(\Pi^\star(\mathcal{K}_u)\right)}_{\text{Sparsity part in Eq. 4}} + \underbrace{w_{\text{concen}} \|\mathbf{F}_{obj}^{\Pi^\star(\mathcal{K}_u)} - \mathbf{F}_u/\|\mathbf{F}_u\| \|_1}_{\text{Concentration part in Eq. 6}} \right), \tag{8}$$

where $w_{\text{class}}, w_{\text{concen}}$ are weight hyper-parameters, and $\tau = 0.8$ is a pre-defined threshold for filtering noisy labels. As verified in Fig. 5, regions with high values in the calculated uncertainty map largely align with areas of noisy segmentation. More analysis can be found in Sec. 4.4.

## 4 EXPERIMENTS

### 4.1 EXPERIMENTS SETTING

**Implementation details.** Our implementation is based on the official codebase of 3D-GS (Kerbl et al., 2023). We utilize the same photometric loss term in Kerbl et al. (2023) to optimize the associated 3D Gaussian parameters. For the Gaussian-level features, we set the feature dimension to 16, following the baseline works such as OmniSeg3D-GS (Ying et al., 2024) and Gaussian Grouping (Ye et al., 2023). To optimize the Gaussian-level features, we apply the same contrastive loss used in Ying et al. (2024). For the object-level codebook, we set the maximum object number $L$ to 256 and use the proposed loss defined in Eq. 8 to optimize the object-level codebook from a random initialization. Empirically, we set $w_{\text{class}} = 1 \times 10^{-3}$ and $w_{\text{concen}} = 1 \times 10^{-1}$ in our experiments. All parameters are jointly optimized, with the number of training iterations set to 30,000 for all datasets. More details are provided in the supplementary material.

**Dataset.** We conduct experiments on the widely-used LERF-Mask dataset (Ye et al., 2023) and the Replica dataset (Straub et al., 2019) to conduct both quantitative and qualitative comparisons. The

LERF-Mask dataset includes three scenes, "figures", "ramen", and "teatime", each with six to ten object segmentation annotations. For the Replica dataset, we select eight scenes for comparisons, where each scene comprises 64 training images and 16 testing images, as processed in Turkulainen et al. (2024). Furthermore, we use the official Segment Anything Model (SAM) (Kirillov et al., 2023) to make predictions and obtain the initial 2D segmentation masks, empirically choosing the largest granularity that provides the object-level segmentation context.

**Metrics.** For the LERF-Mask dataset, we adopt the evaluation protocol from Ye et al. (2023) following the existing works (Ye et al., 2023; Lyu et al., 2024), using the mean Intersection over Union (mIoU) and the boundary IoU (mBIoU) metrics. For the Replica dataset, we first use the linear assignment algorithm to calculate the best matching of IoU between the segmentation predictions and ground-truth data; we then report both the mIoU metric and F-score, using an IoU threshold of 0.5 as the criterion.

**Comparisons.** We compare our proposed method with three types of lifting approaches based on 3D-GS: (i) lifting methods with a preprocessing, such as Gaussian Grouping (Ye et al., 2023) and Gaga (Lyu et al., 2024); (ii) the lifting method with a post-processing, *i.e.*, OmniSeg3D-GS (Ying et al., 2024); and (iii) a direct lifting baseline (*i.e.*, "Panoptic-Lifting-GS" denoted in Tab. 1, Tab. 2, and Tab. 3) that is derived from Panoptic-Lifting (Siddiqui et al., 2023). To ensure fair comparisons, we evaluate the OmniSeg3D-GS baseline using the HDBSCAN (McInnes et al., 2017) algorithm to automatically generate segmentation results. Further, we report metrics under the best-found hyper-parameters, following the common practice used in Bhalgat et al. (2023). Moreover, we benchmark our method against the recent open-vocabulary 3D segmentation techniques, including LERF (Kerr et al., 2023) and Lansplat (Qin et al., 2024).

## 4.2 Main experiments

**LERF-Mask dataset.** To evaluate performance on real-world data, we conduct the experiments using the LERF-Mask dataset (Ye et al., 2023). Quantitative comparisons provided in Tab. 1 demonstrate that our method outperforms all existing lifting methods, as well as open-vocabulary approaches like LERF (Kerr et al., 2023) and Lansplat (Qin et al., 2024). Moreover, visual comparisons between our method and other methods are presented in Fig. 6 (a), demonstrating the effectiveness of our approach in achieving consistent and accurate 3D segmentation. Following the process in the baseline work (Ye et al., 2023), we set the segmentation result to empty if the calculated IoU between predictions and ground truth falls below a predefined threshold.

**Replica dataset.** To further validate the effectiveness of our method, we conduct experiments on the Replica dataset (Straub et al., 2019), which comprises eight distinct scenes. Quantitative comparisons with state-of-the-art methods, presented in Tab. 2, demonstrate that our method achieves the best performances across all metrics. Visual results, illustrated in Fig. 6 (b), further verify that our method not only produces more accurate segmentations for small objects (*e.g.*, vase and button) but also generates significantly fewer artifacts compared to the existing methods. Notably, even when using the optimal hyper-parameters in HDBSCAN (McInnes et al., 2017) for OmniSeg3D-GS (Ying et al., 2024), its post-processing clustering algorithm struggles to balance accuracy for small objects and smooth segmentation for larger objects.

## 4.3 Scalability on varying object numbers

To demonstrate the scalability of our method across varying object quantities, we conduct additional experiments on the widely-used Messy Rooms dataset (Bhalgat et al., 2023), which covers scenes containing up to 500 distinct objects. For fair comparisons, we follow the same evaluation protocol used in the previous work (Bhalgat et al., 2023) to calculate the metric that assesses the consistency of instance IDs across multiple views (Siddiqui et al., 2023), denoting as $PQ^{scene}$ in Tab. 3. Specifically, we choose the segment with largest area in the generated instance segmentation across different views as the background, to generate the binary semantic segmentations for $PQ^{scene}$ metric calculations. This approach avoids the need to optimize an additional semantic feature in our method, as well as in all 3D-GS-based baselines. We compare our method with the 3D-GS-based baselines (*i.e.*, OmniSeg3D-GS and Panoptic-Lifting-GS) and the NeRF-based baselines (*i.e.*, Panoptic Lifting (Siddiqui et al., 2023) and Contrastive Lift (Bhalgat et al., 2023)) for a comprehensive evaluation. As shown in Tab. 3, the quantitative results demonstrate that our method achieves improved performance compared to 3D-GS-based baselines, particularly in scenes with a large number of objects. Moreover, our method achieves results comparable to the current state-of-the-art NeRF-based method (Bhalgat et al., 2023), while requiring significantly less training time.

## 4.4 ABLATION STUDY

We conduct a detailed ablation study to validate the effectiveness of each component in our proposed method. Our baseline solution for codebook learning combines contrastive learning for Gaussian-level features with a cross-entropy loss for the object-level codebook, utilizing the mapping strategy from Siddiqui et al. (2023). Quantitative results presented in Tab. 4 demonstrate that each proposed component significantly enhances our method's performance.

## 4.5 APPLICATIONS

Our method effectively offers an object-level understanding of the 3D scene, which can further facilitate downstream applications. For example, it enables the direct selection of objects in the 3D domain for fundamental copy-and-paste operations. Benefiting from our accurate segmentation results, the edited outputs appear more natural and exhibit fewer artifacts, as illustrated in Fig. 6 (c) left. Furthermore, our method can be easily extended to provide multi-granularity understanding, by simply employing segmentation at various granularities (*e.g.*, three-level granularity for SAM). This capability enables end-to-end multi-scale object selections, as showcased in Fig. 6 (c) right.

Table 1: Quantitative comparisons of segmentation quality on the LERF-Mask dataset (Ye et al., 2023). We report the mIoU and mBIoU metrics following Gaussian Grouping (Ye et al., 2023). * indicates self-implementation, and † indicates that the results are reported under the best-found hyper-parameter, *i.e.*, minimal cluster size for HDBSCAN (McInnes et al., 2017).

| Method | Venue | mIoU(%) | mBIoU (%) |
|---|---|---|---|
| LERF | ICCV'23 | 37.2 | 29.3 |
| LangSplat | CVPR'24 | 57.6 | 53.6 |
| Gaussian Grouping | ECCV'24 | 72.8 | 67.6 |
| Gaga | Arxiv'24 | 74.7 | 72.2 |
| OmniSeg3D-GS (†) | CVPR'24 | 74.7 | 71.8 |
| Panoptic-Lifting-GS | * | 70.7 | 65.8 |
| Ours | - | 80.9 | 77.1 |

Table 2: Quantitative comparisons of segmentation quality on the Replica dataset (Straub et al., 2019). We report the mIoU, and F-score metrics. * indicates self-implementation, and † indicates that the results are reported under the best-found hyper-parameter, *i.e.*, minimal cluster size for HDBSCAN (McInnes et al., 2017).

| Method | mIoU(%) | F-score (%) |
|---|---|---|
| Gaussian Grouping | 23.6 | 30.4 |
| OmniSeg3D-GS (†) | 39.1 | 35.9 |
| Panoptic-Lifting-GS (*) | 25.3 | 32.9 |
| Our | 41.6 | 43.9 |

Table 3: Results on the Messy Rooms dataset (Bhalgat et al., 2023). Following Bhalgat et al. (2023), $PQ^{scene}$ metric is reported on both the "old room" and "large corridor" environments with an increasing number of objects in the scene $(25, 50, 100, 500)$. Note that, we test the training time for all methods using a single NVIDIA 3090 RTX GPU.

| Type | Method/ Number | Old Room Environment (%) | | | | Large Corridor Environment(%) | | | | Mean(%) | Training (h) |
|---|---|---|---|---|---|---|---|---|---|---|---|
| | | 25 | 50 | 100 | 500 | 25 | 50 | 100 | 500 | | |
| NeRF | Panoptic Lifting | 73.2 | 69.9 | 64.3 | 51.0 | 65.5 | 71.0 | 61.8 | 49.0 | 63.2 | $\geq 20$ |
| | Contrastive Lift | 78.9 | 75.8 | 69.1 | 55.0 | 76.5 | 75.5 | 68.7 | 52.5 | 69.0 | $\geq 20$ |
| GS | Panoptic-Lifting-GS (*) | 67.5 | 65.1 | 59.4 | 46.1 | 62.2 | 65.3 | 57.5 | 45.5 | 58.6 | $\approx 1$ |
| | OmniSeg3D-GS (†) | 80.1 | 72.4 | 61.4 | 46.8 | 74.9 | 79.6 | 63.9 | 48.5 | 66.0 | $\approx 1$ |
| | Ours | 79.1 | 72.2 | 65.9 | 53.9 | 77.0 | 78.9 | 70.7 | 54.1 | 69.0 | $\approx 1$ |

Table 4: Ablation study for the proposed components.

| Method | mIoU(%) | F-score (%) |
|---|---|---|
| Baseline solution (w/ codebook) | 29.5 | 39.2 |
| + concentration term in association learning module | 36.3 | 41.3 |
| + area-aware ID mapping in association learning module | 39.2 | 41.0 |
| + noisy label filtering (full method) | 41.6 | 43.9 |

## 5 CONCLUSION

We propose a new unified *object-aware lifting* approach based on 3D-GS for constructing accurate and efficient 3D scene segmentations. Specifically, we introduce a novel object-level codebook to incorporating an explicit object-level understanding of the 3D scene by learning a representation for each object. Method-wise, we first augment each Gaussian point with a Gaussian-level point

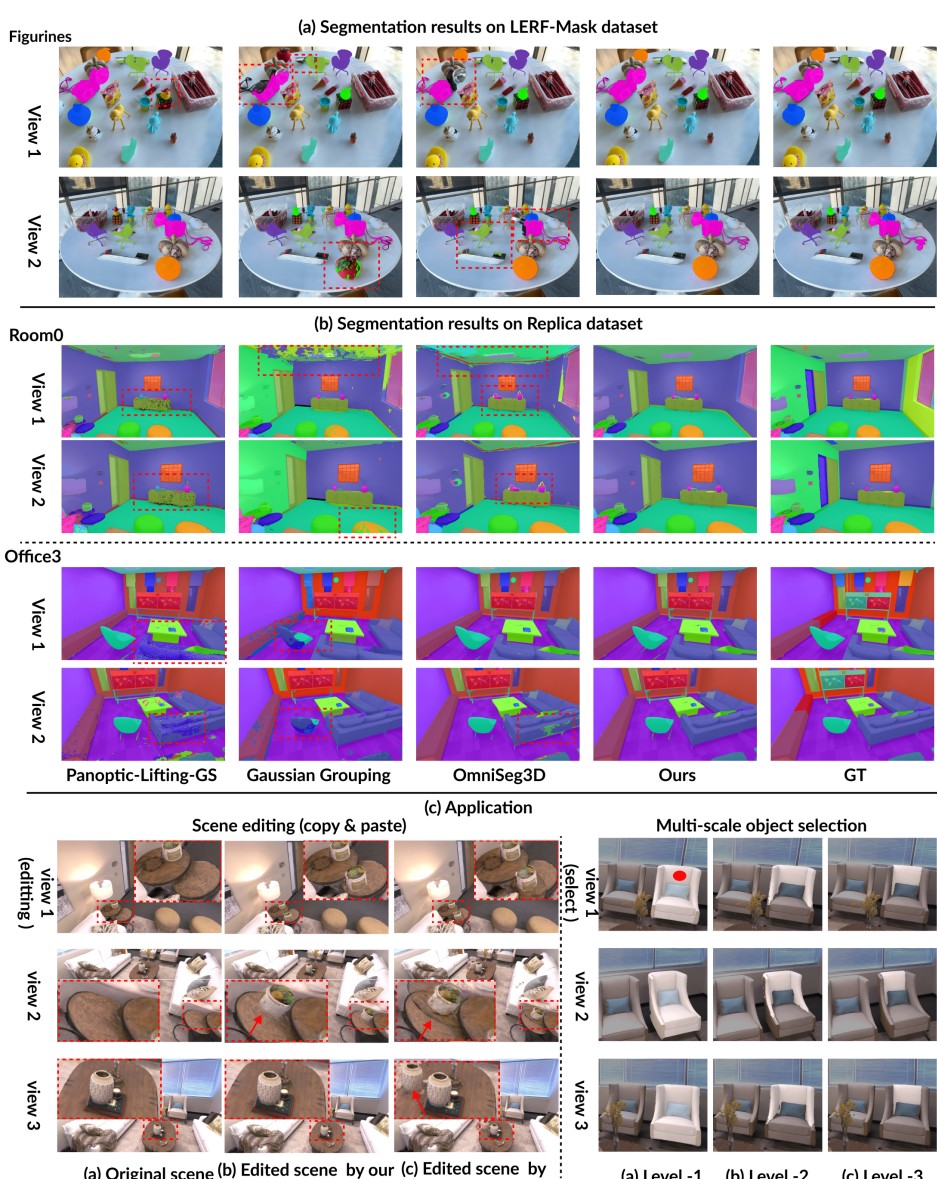

Figure 6: Qualitative comparison of our method with previous methods. We provide visual comparisons on the LERF-Masked dataset (Ye et al., 2023) in (a); and on the Replica dataset (Straub et al., 2019) in (b). Moreover, we present the application results in (c). As shown in the left part of (c), we select the potted plants in view 1 and apply the copy & paste operations to the associated Gaussian points. The consistent editing results in view 2 and view further demonstrate the advantages of our method. In contrast, using segmentations derived from Gaussian Grouping (Ye et al., 2023) leads to severe artifacts and can even adversely affect unrelated object such as the vase observed in view 3. In addition, we illustrate the multi-scale object selection application in the right part of (c). By clicking on the red point in view 1, we consistently select the sofa instance at three different granularities across multiple views.

feature and adopt the contrastive loss to optimize these features. Then, we formulate the object-level codebook representation and associate it with the Gaussian-level features for object-aware segmentation prediction. To ensure effective and robust learning for the object-level codebook, we further propose the association learning module and the noisy label filtering module. Extensive experimental results manifest the effectiveness of our method over the state of the arts. Further analysis on the Messy Rooms dataset also shows its scalability in handling large numbers of objects.

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
