# OBJECT-AWARE LIFTING FOR 3D SCENE SEGMENTATION IN GAUSSIAN SPLATTING

In this supplementary material, we provide more implementation details (Sec. A), more quantitative and qualitative comparisons (Sec. B), and more detailed analysis (Sec. C) about our proposed method.

## A    IMPLEMENTATIONS DETAILS

We implement our method based on the 3D Gaussian Splatting (3D-GS) Kerbl et al. (2023) representation. Specifically, we set the dimension of the Gaussian-level feature as 16 to maintain a fair comparison with previous baselines (*e.g.*, Gaussian Grouping (Ye et al., 2023) and OmniSeg3D-GS (Ying et al., 2024)). For the learning of original properties in 3D Gaussian points, we use the same learning rate and same density control as the in the original work (Kerbl et al., 2023). For the Gaussian-level feature, we utilize the Adam optimizer with a learning rate of 0.0025. For the object-level code, we employ the Adam optimizer with a learning rate of 0.0005. We jointly train all parameters for 30,000 iterations on each dataset covered in this work, using a single NVIDIA RTX 3090.

## B    MORE QUANTITATIVE AND QUALITATIVE COMPARISONS

**Quantitative comparisons with OmniSeg3D-GS.**    Since OmniSeg3D-GS (Ying et al., 2024) only learns feature embeddings, we equip OmniSeg3D-GS with HDBSCAN clustering algorithm (McInnes et al., 2017) to produce the final segmentation results. We report the performance under the optimal best-found hyper-parameter (i.e., minimal cluster size) for HDBSCAN, following the same strategy used in Contrastive Lift (Bhalgat et al., 2023). Specifically, we utilize the training views to search for the best hyper-parameter for each scene, setting the search range from 10 to 200, as suggested in "Tuning Clustering Hyperparameter" (Bhalgat et al., 2023). As shown in Fig. 1, while the exhaustive search can improve performance, it is still behind our method, which achieves consistent results without the need for hyperparameter tuning.

**Qualitative comparisons.**    In addition to the visual results presented in the main paper, we provide more qualitative comparisons in Fig. 2, Fig. 3 and Fig. 4. These visual results further demonstrate that our method delivers more accurate and consistent segmentation across various views, while also minimizing artifacts.

## C    MORE ANALYSIS OF PROPOSED COMPONENTS

Table 1: Effectiveness analysis of the proposed area-aware ID mapping method. We compare the segmentation results of pseudo-labels generated by our area-aware ID mapping and the approach proposed in Panoptic Lifting (Siddiqui et al., 2023).

| ID matching strategy | mIoU(%) | F-score(%) |
|---|---|---|
| Siddiqui et al. (2023) | 30.3 | 30.4 |
| Proposed area-aware ID mapping | 31.7 | 33.5 |

**Area-aware ID mapping.**    To further verify the effectiveness of our area-aware ID mapping, we present additional quantitative comparisons between the generated pseudo-labels by our area-aware

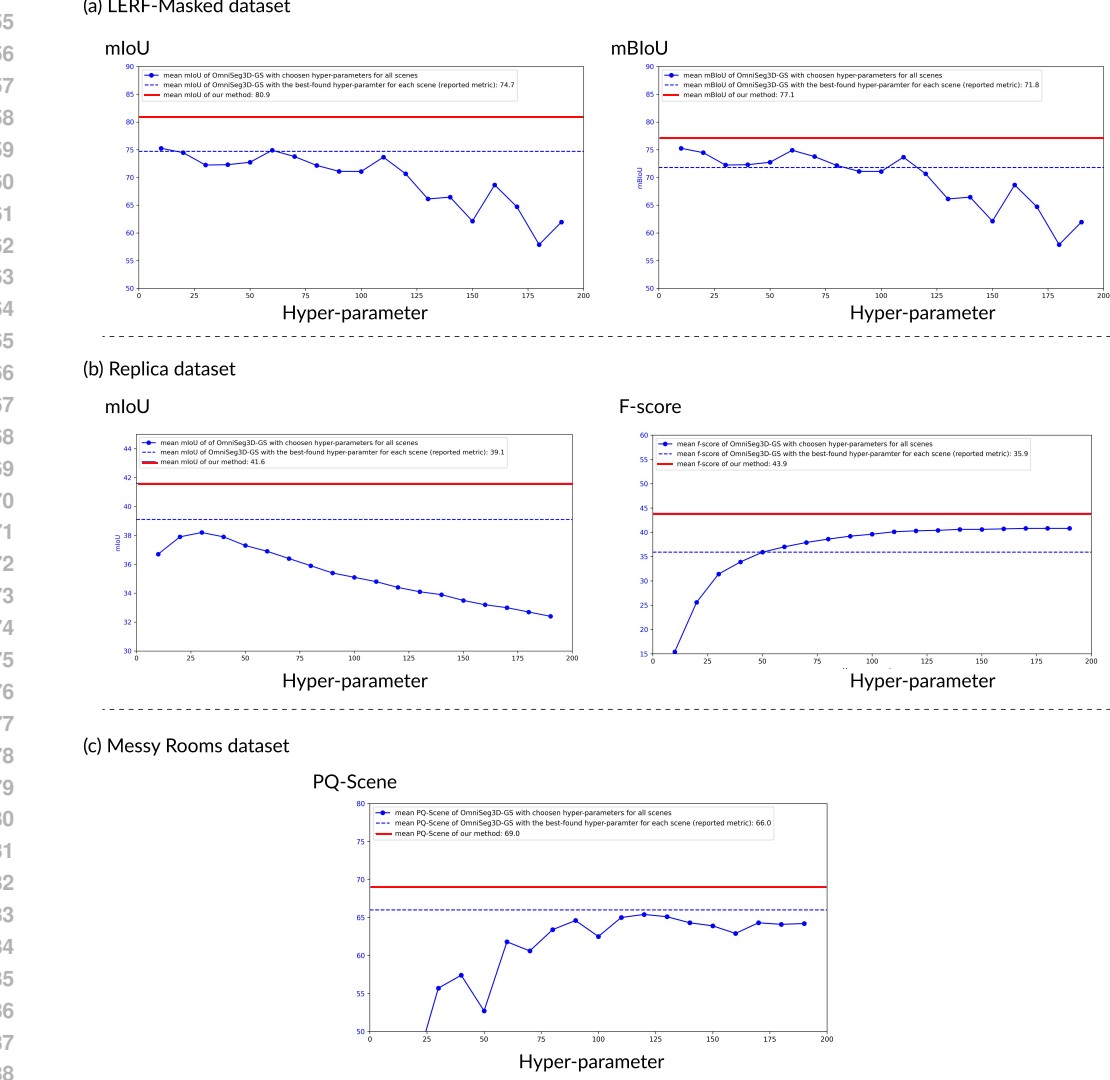

Figure 1: The detailed comparison between our method and OmniSeg3D-GS methods on the LERF-Mask dataset (Ye et al., 2023), Replica dataset (Straub et al., 2019) and the Messy Rooms dataset (Bhalgat et al., 2023) dataset. For LERF and Replica datasets, we utilize the mIoU metric to search the best hyper-parameter for each scene. For the Messy Rooms dataset, we utilize the PQ-Scne to select the best hyper-parameter for each scene. Note that our method is denoted by the red color, and OmniSeg3D-GS is denoted by the blue color.

Table 2: Quantitative comparisons of using different thresholds $\tau$ values in the noisy label filtering module.

| $\tau$ | 0.75 | 0.80 (default) | 0.85 |
|---|---|---|---|
| mIoU(%) | 40.0 | 41.6 | 40.4 |
| F-score(%) | 43.0 | 43.9 | 43.7 |

ID mapping method and the method proposed in Panoptic Lifting (Siddiqui et al., 2023). The results shown in Tab. 1 verify that the pseudo-labels generated by our area-aware ID mapping are more accurate and consistent.

**Sensitivity to different per-defined values in noisy label filtering.** We investigate the impact of varying the predefined threshold used to filter noisy labels in the noisy label filtering module. In

Table 3: Ablation study on the effectiveness of our gradient-blocking design.

| Method | mIoU(%) | F-score (%) |
|---|---|---|
| Full model | 41.6 | 43.9 |
| Full model w/o gradient-blocking design | 39.7 | 39.8 |

our main experiments, we set a predefined threshold of $\tau = 0.8$ to filter noisy segmentations in the noisy label filtering module. To investigate the impact of this threshold, we conduct additional experiments using two different values ($\tau = 0.75$ and $0.85$). As shown in Tab. 2, the results remain rather stable despite moderate changes in the threshold $\tau$.

**Gradient-blocking.** In practice, we block the gradient derived from the association constraints from propagating to the Gaussian-level features. This gradient-blocking design ensures that the Gaussian-level features are exclusively optimized through the contrastive loss. The ablation study in Tab. 3 demonstrates that this design improves optimization stability.

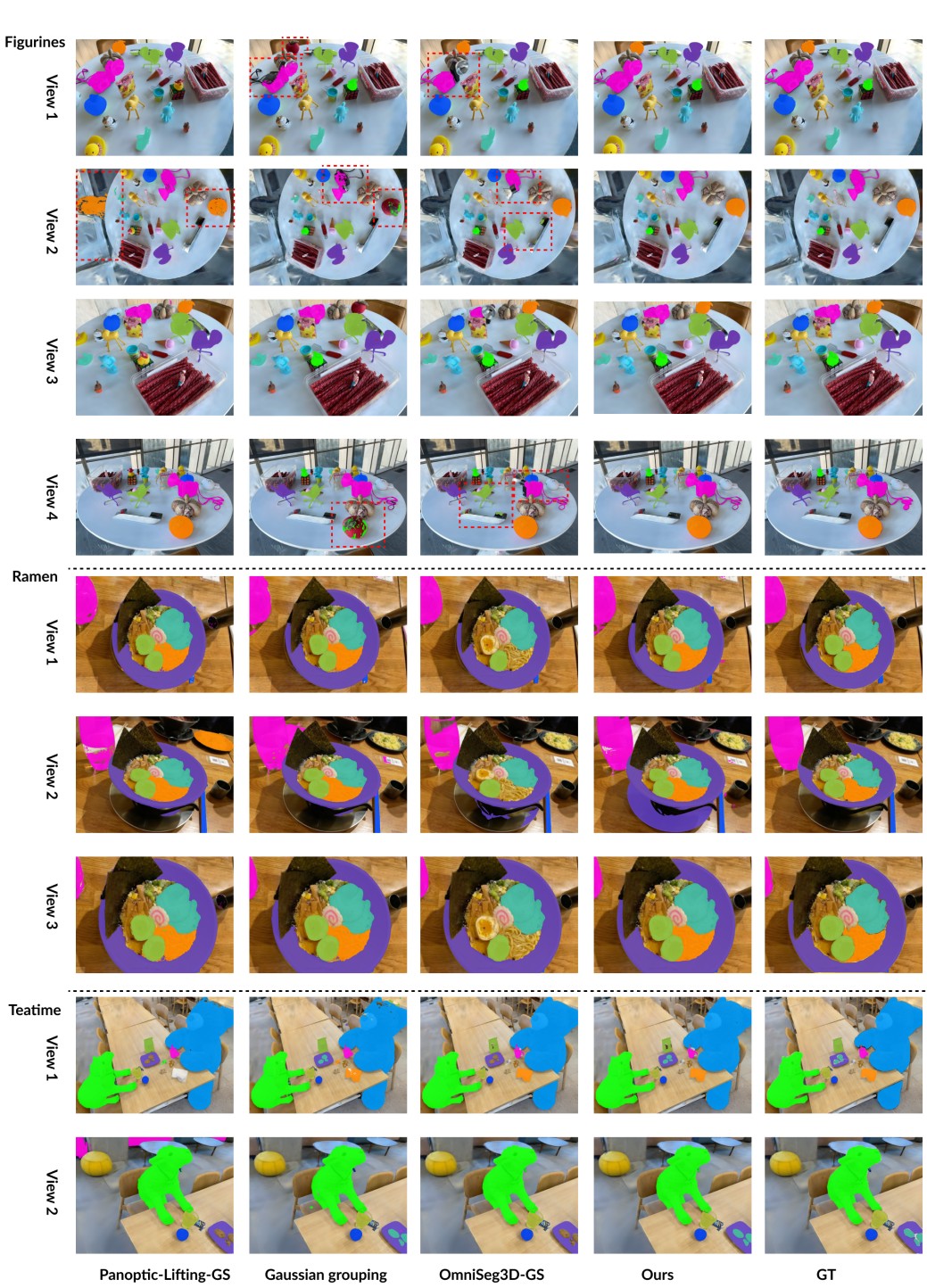

Figure 2: Visual comparisons between our method and previous methods on the LERF-Masked dataset (Ye et al., 2023).

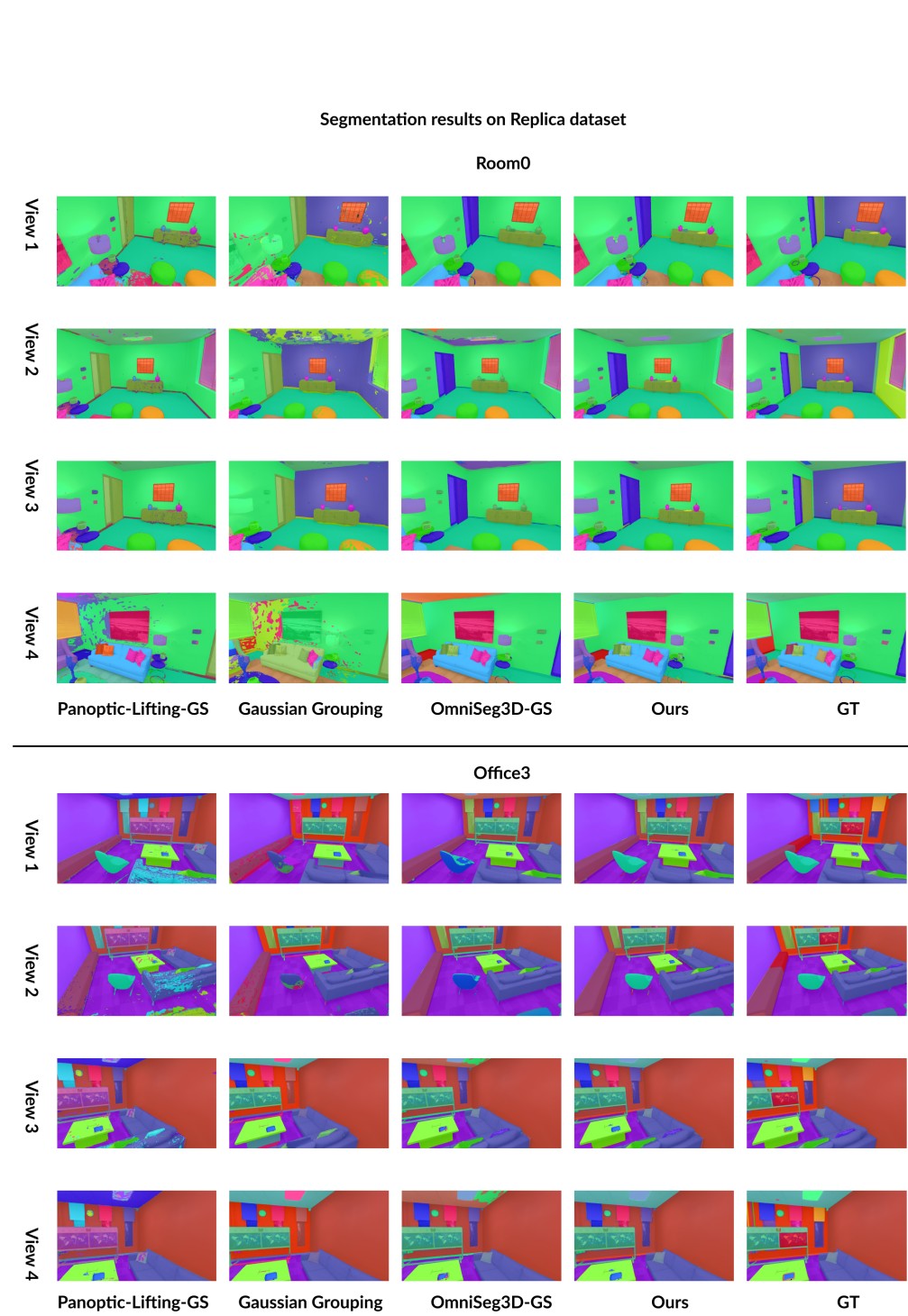

Figure 3: Visual comparisons between our method and previous methods on the Replica dataset (Straub et al., 2019).

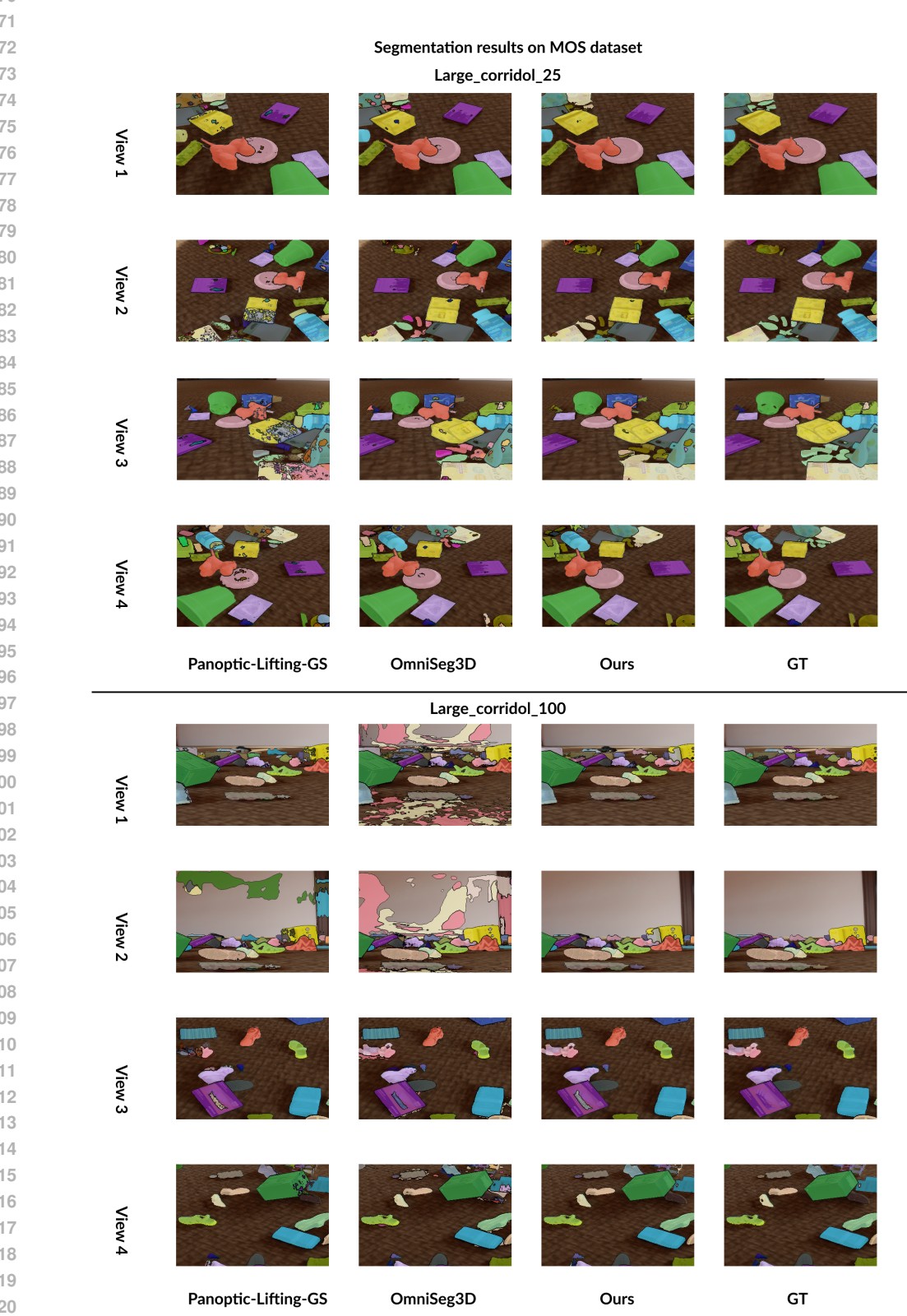

Figure 4: Visual comparisons between our method and previous methods on the Messy Rooms dataset (Straub et al., 2019).