# OpenReview forum: "Object-aware lifting for 3D scene  segmentation in Gaussian splatting"
_ICLR.cc/2025/Conference — ICLR 2025 Conference Withdrawn Submission_

### Official Review · Reviewer_A1ye · 2024-11-01

**Soundness:** 2
**Presentation:** 3
**Contribution:** 2
**Rating:** 3
**Confidence:** 5

**Summary:**

While this paper proposes a 3D object segmentation method using a 3D Gaussian Splatting (3D-GS) framework to align 2D Segment Anything Model (SAM) masks into 3D space, the approach ultimately fails to adequately address critical issues that impact its potential contribution to the field.

**Strengths:**

The paper is clearly structured, presenting its methodology in a logical progression. It introduces an object-level codebook and supplementary modules aimed at eliminating clustering post-processing steps, which are known to be hyperparameter-sensitive. Additionally, the paper includes experimental results on three datasets, LERF-Masked, Replica, and Messy Rooms, demonstrating some improvements in segmentation consistency and computational efficiency within the scope of these datasets over selected baselines.

**Weaknesses:**

1. Lack of Clarity and Detail: Key aspects of the method are inadequately explained, notably the association learning and noisy label filtering modules, which are limited to brief descriptions. This lack of detail impedes a clear understanding of the method’s workings and its practical applicability. Furthermore, there is no discussion on whether image constraints were utilized for reconstruction, nor are any re-rendering metrics presented to assess quality, limiting confidence in the segmentation outcomes.
2. Motivation and Novelty: While the paper introduces an association learning module and noisy label filtering module, these elements lack significant novelty in the broader context of 3D segmentation. Similar techniques, such as object-level codebooks, have already been leveraged by prior work, including Object-NeRF and DM-NeRF, which are not sufficiently discussed. Without a thorough comparison with these methods, it remains unclear how this approach substantially advances beyond existing techniques.
3. Technical Complexity and Innovation: The technical contributions are minimal. The association learning module and noisy label filtering module employ approaches already prevalent in segmentation literature. The sole modifications appear in Equations (6) and (7), with limited impact demonstrated in the experimental results (Table 3) compared to baseline methods like Panoptic Lifting and Contrastive Lift. This limited innovation detracts from the potential significance of the proposed method.
4. Incomplete Experimental Validation: The paper lacks key experiments necessary to assess the proposed method’s effectiveness. Notably, NeRF-based lifting methods were not evaluated on the Replica and LERF-Masked datasets, which limits the comparison of this approach with established baselines. Furthermore, the robustness of segmentation across different SAM mask granularities is unexamined, leaving uncertainty regarding the method’s adaptability to variations in segmentation granularity. Essential metrics, such as F-score calculations in 2D versus 3D point clouds and segmentation accuracy across different viewing angles, are also absent. These omissions hinder a comprehensive understanding of the method's applicability across diverse scenes and conditions. Additionally, Table 4 provides a breakdown of comparative experiments on various components of the design, yet it does not specify which dataset these results were obtained from, further limiting the interpretability of the findings.

**Questions:**

1. Could you clarify whether the F-Score reported in the paper is calculated on the 2D segmentation maps or on the 3D point clouds?
2. Beyond the copy-and-paste operations for scene editing, does your method support rigid transformations or deformation operations on segmented objects? Additionally, are you able to validate the re-rendered scenes with quantitative comparisons and assess the geometric accuracy of these edits?
3. In Figure 6(b), View 3 appears to show the original image. Could you provide additional segmentation quality results from larger viewing angles or greater distances from the initial position? This would help assess whether the method can accurately segment Gaussian spheres, especially along object edges.
4. It appears that only 64 images were used as training data for the Replica dataset. If this is correct, could you explain the rationale behind choosing this limited dataset size?
5. In Table 4, comparative experiments are provided for different components of your design. However, the dataset used for these results is not specified. Could you clarify which dataset was used for the experiments in Table 4?

---

### Official Review · Reviewer_beAN · 2024-11-02

**Soundness:** 3
**Presentation:** 4
**Contribution:** 3
**Rating:** 6
**Confidence:** 5

**Summary:**

Accurate 3D scene segmentation enhances scene understanding and scene editing tasks, unlocking numerous downstream applications in AR/VR, autonomous vehicles, and robotics. Lifting features from 2D instance/semantic segmentation models to recent 3D representations such as NeRF and Gaussian Splatting (3DGS) is a popular technique for performing 3D scene segmentation.

Existing state-of-the-art lifting methods have the following issues:
- Works like Panoptic-Lifting use linear assignment + classification loss to learn 3D segmentations, but these learned representations lack semantic meaningfulness (L42-44)
- Works like Gaussian Grouping and GAGA utilize association techniques as a preprocessing step for view consistency. However, the preprocessing step can produce inaccurate results (L46-47).
- Works like Contrastive-Lift do not require preprocessing steps and encode instance information in a feature field, which is optimized using contrastive losses. However, they require a post-processing step like HDBSCAN to predict the final instance segmentation masks. (L48-49)

This work proposes a unified lifting framework for 3D instance segmentation, which does not require any preprocessing or post-processing steps. Contributions of this work are summarized as follows:
- They propose a unified framework for accurate 3D instance segmentation by introducing an object-level codebook representation.
- To train the proposed codebook effectively, they present novel association learning and noise filtering modules.
- The proposed method achieves SOTA on public benchmarks.

**Strengths:**

1. **Paper-Writing and Presentation**: The overall quality of the paper, including the clarity of writing and the presentation of figures, was excellent. The well-structured format and coherent flow made the paper easy to read and navigate.

2. **Clear Motivation and Shortcomings of the existing methods**: The authors provide a clear and thorough explanation of the limitations of current methods, as detailed in L40-53 and illustrated in Figure 1. This explanation establishes a strong context and rationale for the proposed problem statement.

3. **Technical novelty**: The proposed object-level codebook represents an innovative technical contribution that reduces the reliance on post-processing steps. To enable effective training of this codebook, the authors introduce three key components:  area-aware ID mapping, a concentration term, and a noisy label filtering module. These enhancements improve the performance of the final method as described in Ablation studies in Table 4. The significance of these contributions is substantial, and their potential impact warrants sharing them with the broader research community to advance the field.

4. **Beats SOTA on benchmark datasets**: The authors evaluate their proposed method using three well-known datasets: Replica, LeRF, and Messy Rooms. Across all these benchmarks, the method demonstrates state-of-the-art (SOTA) performance, underscoring its effectiveness. Additionally, qualitative results highlight the method's ability to maintain multi-view consistency.

**Weaknesses:**

1. **Training time comparison with SOTA methods such as Gaussian Grouping**: The authors discuss training time in Table 3; however, a comparison with the Gaussian-Grouping method is not included. To further improve the quality of the manuscript, the authors should provide a detailed breakdown of the training time for their proposed method.

2. **Inconsistency in the metrics used in the paper**: $PQ^{scene}$ is a scene-level extension of standard Panoptic Quality (PQ) that takes into account the consistency of instance IDs across views/frames (aka tracking). This metric is reported only for the Messy Rooms dataset and is not provided for the LERF-Mask or Replica datasets. I would recommend that the authors address this during the rebuttal phase.

3. **Missing Results on novel-view synthesis task**: It is essential to report PSNR, SSIM and LPIPS for the novel-view synthesis task for standard datasets. Check Table 1 in the Gaussian Grouping paper.

4. **Missing Comparison on Scannet scenes**: Contrastive-Lift and Gaussian-Grouping report the results on the Scannet dataset as well. For the thoroughness of the experiments, I recommend that the authors address this during the rebuttal phase.

5. **Robustness to the choice of instance segmentation method**: Currently, the proposed method employs Segment Anything Model (SAM). However, how is the performance affected when a different segmentation model, such as MaskFormer, is used? Is this a limitation of the proposed method?

**Questions:**

1. L42-44: Can authors clarify what they mean by "lacks semantically meaningful instance features"?

2. L236: Can authors clarify what is the cause of this accumulated error?

3. Typo in Eq. 2. It should be L instead of L-1 as the id of the first element starts from 1.

4. Comparison on Messy Rooms Dataset: Please share the results for the Gaussian-Grouping method in Table 3.

---

### Official Review · Reviewer_tvsX · 2024-11-03

**Soundness:** 2
**Presentation:** 3
**Contribution:** 2
**Rating:** 5
**Confidence:** 5

**Summary:**

- The paper tries to solve the task of 3D scene segmentation using 2D multiview instance masks within a single method,  avoiding any additional pre-processing or post-processing step.
- The method suggested by the author uses per-Gaussian level features, object level features and later use association functions to bring in mutual correlation between these features.
- The authors also propose a noisy label filtering module by estimating an uncertainty map for the segmentation masks.
- To show the effectiveness and scalability of the proposed method, the authors have reported quantitative numbers on popular datasets including LeRF-mask dataset and Replica and Messy rooms

**Strengths:**

- The gaussians in the proposed method get object-level understanding of the 3D scene by learning from the Gaussian-level features, rather than utilizing post-processing techniques like clustering which requires parameter tuning.
- Compared to other methods, the proposed pipeline claims to not overlook small objects.
- The paper is easy to follow.

**Weaknesses:**

- The paper does not discuss regarding the extra memory overhead that would incur by introducing separate set of features at gaussian level and object level.
- The quantitative numbers in some of the tables seem spurious to me. For example the authors claim the mIou value of Gaussian grouping in replica is 23.6 but the original paper has reported 71.15. The authors have not reported the mIOU numbers of Contrastive Lift, which has reported 67.0 mIOU for replica.
- In Fig4 the authors claim that their method produces pseudo labels that are more view consistent to facilitate the codebook learning, but the pillows on the sofas do not have consistent masks. This is also against the claims that the proposed method does good on small objects in the scene.
- There are a bunch of typo and errors that needs to be addressed in the manuscript Example: Inconsistent tick mark in Fig 1 (c),
Fig1 (“when” —> “where”).

**Questions:**

- The authors claim that other methods lack semantically meaningful instance features. What would happen if we used L-Seg features or clip features instead of the codebooks for understanding the object features? Won't it help us query objects in the scene and improve interactivity? [Not a weakness just a question]
 - How much extra time overhead does clustering based methods take? Is the proposed method saving significant time by avoiding pre-processing/ post-processing methods apart from finding hyper-parameters?
- The significant improvement in training time is mostly because of using 3DGS as a base representation. Do you think components from your pipeline can be used to improve existing SOTA NeRF based methods in similar tasks?

Please correct me if you think I have misunderstood any aspect of the paper and address the Weakness section.

---

### Official Review · Reviewer_ce3Y · 2024-11-03

**Soundness:** 3
**Presentation:** 3
**Contribution:** 2
**Rating:** 5
**Confidence:** 5

**Summary:**

This work proposes a framework for an object-aware segmentation mask lifting method for 3D Gaussian Splatting. A novel learnable object-level codebook is introduced to account for objects in the 3D scene, enabling explicit object-level understanding. Similar to previous works, a contrastive loss is also applied to regress a feature field for each Gaussian. With the codebook formulation, the encoded object-level features are associated with Gaussian-level point features for segmentation predictions. Experiments are conducted on the LERF-Masked, Replica, and Messy Rooms datasets for evaluation.

**Strengths:**

1. Introduced a codebook formulation for 3D Gaussian Splatting segmentation lifting.

2. Both qualitative and quantitative results demonstrate the effectiveness of this work.

**Weaknesses:**

1. Although the authors use the word "novel" multiple times in the abstract, the proposed work seems incremental when compared to previous clustering-based methods, such as Gaussian-grouping [1].

2. As the main goal of this work is to lift 2D masks to masks for each 3D Gaussian, results on object removal in 3D scenes should be presented. Only projecting 3D segmentation masks back into 2D images is not convincing.

3. More qualitative comparison with Gaussian-grouping [1] should be conducted, especially for downstream tasks, such as 3D removal, inpainting, and editing.

4. Despite tremendous work on 3D Gaussian Splatting Segmentation, such as Gaga [2], Click-Gaussian [3], and FlashSplat [4], discussion on these recent works is still necessary to justify the setting of this work. In particular, FlashSplat highlighted that each 3D Gaussian may have multiple semantic labels, as they are shared between objects in rendering. In such a setting, the per-Gaussian features can be ambiguous, and enforcing each Gaussian to have only one semantic label in this work would be inherently unreasonable.



[1] "Gaussian grouping: Segment and edit anything in 3d scenes." ECCV 2024

[2]  "Gaga : Group Any Gaussians via 3D-aware Memory Bank" Arxiv 2024

[3] "Click-Gaussian: Interactive Segmentation to Any 3D Gaussians" ECCV 2024

[4] "FlashSplat: 2D to 3D Gaussian Splatting Segmentation Solved Optimally" ECCV 2024

**Questions:**

please refer to the weakness part.

---

### Note · Authors · 2024-11-13

I have read and agree with the venue's withdrawal policy on behalf of myself and my co-authors.